# Thrifty wide-context models of B cell receptor somatic hypermutation

Kevin Sung[1], Mackenzie M Johnson[1], Will Dumm[1], Noah Simon[2], Hugh Haddox[1], Julia Fukuyama[3], Frederick A Matsen[4,5,6]*

[1]Computational Biology Program, Fred Hutchinson Cancer Center, Seattle, United States; [2]Department of Biostatistics, University of Washington, Seattle, United States; [3]Department of Statistics, Indiana University, Bloomington, United States; [4]Howard Hughes Medical Institute, Seattle, United States; [5]Department of Genome Sciences, University of Washington, Seattle, United States; [6]Department of Statistics, University of Washington, Seattle, United States

## eLife Assessment

This study provides an **important** method to model the statistical biases of hypermutations during the affinity maturation of antibodies. The authors show **convincingly** that their model outperforms previous methods with fewer parameters; this is made possible by the use of machine learning to expand the context dependence of the mutation bias. They also show that models learned from nonsynonymous mutations and from out-of-frame sequences are different, prompting new questions about germinal center function. Strengths of the study include an open-access tool for using the model, a careful curation of existing data sets, and a rigorous benchmark; it is also shown that current machine-learning methods are currently limited by the availability of data, which explains the only modest gain in model performance afforded by modern machine learning.

*For correspondence:
matsen@fredhutch.org

**Competing interest:** The authors declare that no competing interests exist.

**Abstract** Somatic hypermutation (SHM) is the diversity-generating process in antibody affinity maturation. Probabilistic models of SHM are needed for analyzing rare mutations, understanding the selective forces guiding affinity maturation, and understanding the underlying biochemical process. High-throughput data offers the potential to develop and fit models of SHM on relevant data sets. In this article, we model SHM using modern frameworks. We are motivated by recent work suggesting the importance of a wider context for SHM; however, assigning an independent rate to each k-mer leads to an exponential proliferation of parameters. Thus, using convolutions on 3-mer embeddings, we develop 'thrifty' models of SHM of various sizes; these can have fewer free parameters than a 5-mer model and yet have a significantly wider context. These offer a slight performance improvement over a 5-mer model, and other modern model elaborations worsen performance. We also find that a per-site effect is not necessary to explain SHM patterns given nucleotide context. Also, the two current methods for fitting an SHM model—on out-of-frame sequence data and on synonymous mutations—produce significantly different results, and augmenting out-of-frame data with synonymous mutations does not aid out-of-sample performance.

## Introduction

Antibodies are an essential component of the adaptive immune response. They are secreted by B cells, and when displayed on the surface of B cells are called B cell receptors. When stimulated by antigen binding, B cells undergo a process called 'affinity maturation' which involves mutation and selection. The mutation process happens at a very high rate relative to the rate of normal somatic mutation

and is called somatic hypermutation (SHM). As such, it is an essential part of the adaptive immune response. It is generated by a complex collection of interacting pathways of DNA damage and error-prone repair, which have been elucidated through decades of research (*Wagner and Neuberger, 1996*; *Teng and Papavasiliou, 2007*; *Methot and Di Noia, 2017*; *Pilzecker and Jacobs, 2019*). These pathways lead to a very non-uniform distribution of mutations.

Furthermore, the mutation biases are predictable from local sequence context. Many papers have investigated predictors of mutation rates from molecular sequence, including early work establishing the biases (*Dunn-Walters et al., 1998*; *Rogozin and Kolchanov, 1992*; *Rogozin and Diaz, 2004*), to parametric models estimating the mutability based on local sequence 'motif', or sequence neighborhood around a focal base (*Yaari et al., 2013*; *Elhanati et al., 2015*; *Cui et al., 2016*; *Feng et al., 2019*; *Fisher et al., 2025*).

Such models are important when predicting the probability of amino acid changes in affinity maturation, for example, for understanding the prospects of selecting such mutations in reverse vaccinology (*Martin Beem et al., 2023*; *Wiehe et al., 2018*) or in computing a model of natural selection on antibodies (*McCoy et al., 2015*; *Hoehn et al., 2017*; *Hoehn et al., 2019*).

The most popular models for SHM are the S5F 5-mer model and its variants (*Yaari et al., 2013*; *Cui et al., 2016*). They have shown their worth for over a decade now, including tasks such as predicting the probability of mutations to mature broadly neutralizing antibodies against HIV (*Wiehe et al., 2018*; *Wiehe et al., 2022*). However, biological considerations suggest that a wider context should be considered.

Indeed, the consensus view of SHM requires processes such as patch removal around a lesion created by the activation-induced cytidine deaminase (AID) enzyme (*Pilzecker and Jacobs, 2019*) and subsequent error-prone repair. Thus, for example, the presence of an AID hotspot several bases away may influence the probability of a mutation at a focal base. More recently, mesoscale-level sequence effects on AID deamination potentially deriving from local DNA sequence flexibility have been discovered (*Wang et al., 2023*). In addition, other work has found that position in the sequence can influence SHM (*Cohen et al., 2011*; *Spisak et al., 2020*; *Zhou and Kleinstein, 2020*).

This begs the question of how one could use more complex models to predict SHM. 7-mer models, which have three flanking bases on either side of the focal base, have been used (*Elhanati et al., 2015*; *Marcou et al., 2018*). However, one cannot simply increase the size of the k-mer model indefinitely because the number of parameters grows exponentially with the size of the k-mer. More recent models of SHM include position-specific terms (*Spisak et al., 2020*) and context models of size up to 21 parameterized by a convolutional neural network (*Tang et al., 2022*). In other contexts, models based on the transformer architecture, *Vaswani et al., 2017* have shown great success, raising the question of whether such an architecture could be used here.

In this article, we develop new models using modern frameworks and provide a comprehensive evaluation of the models. We especially focus on the development of parameter-efficient convolutional neural networks of various sizes, which we call 'thrifty' models. These models have wide nucleotide context yet can have fewer parameters than a 5-mer model, while providing slightly better performance on metrics in train and test time. On the other hand, we find that elaborations such as a per-site rate and transformer only harm out-of-sample performance. We also find a clear difference between training models to predict well on out-of-frame data, compared to training models to predict well on synonymous mutations. To make these models useful for the community, we have released an open-source Python package https://github.com/matsengrp/netam (*Matsen and Dumm, 2025*), with pretrained models and a simple API. Our analysis for this article is reproducible via https://github.com/matsengrp/thrifty-experiments-1 (copy archived at *Sung, 2025*).

## Results
### Overview of data preparation and objective

We will begin with an overview of our models and data. Full details are provided in 'Materials and methods'.

Our objective in this project is to predict the probability of observed SHM in a child sequence relative to a parent sequence. We follow previous work (*Spisak et al., 2020*) in overall goal and data setup. Specifically, we predict mutations in BCR sequences that are out-of-frame, that is, such that the

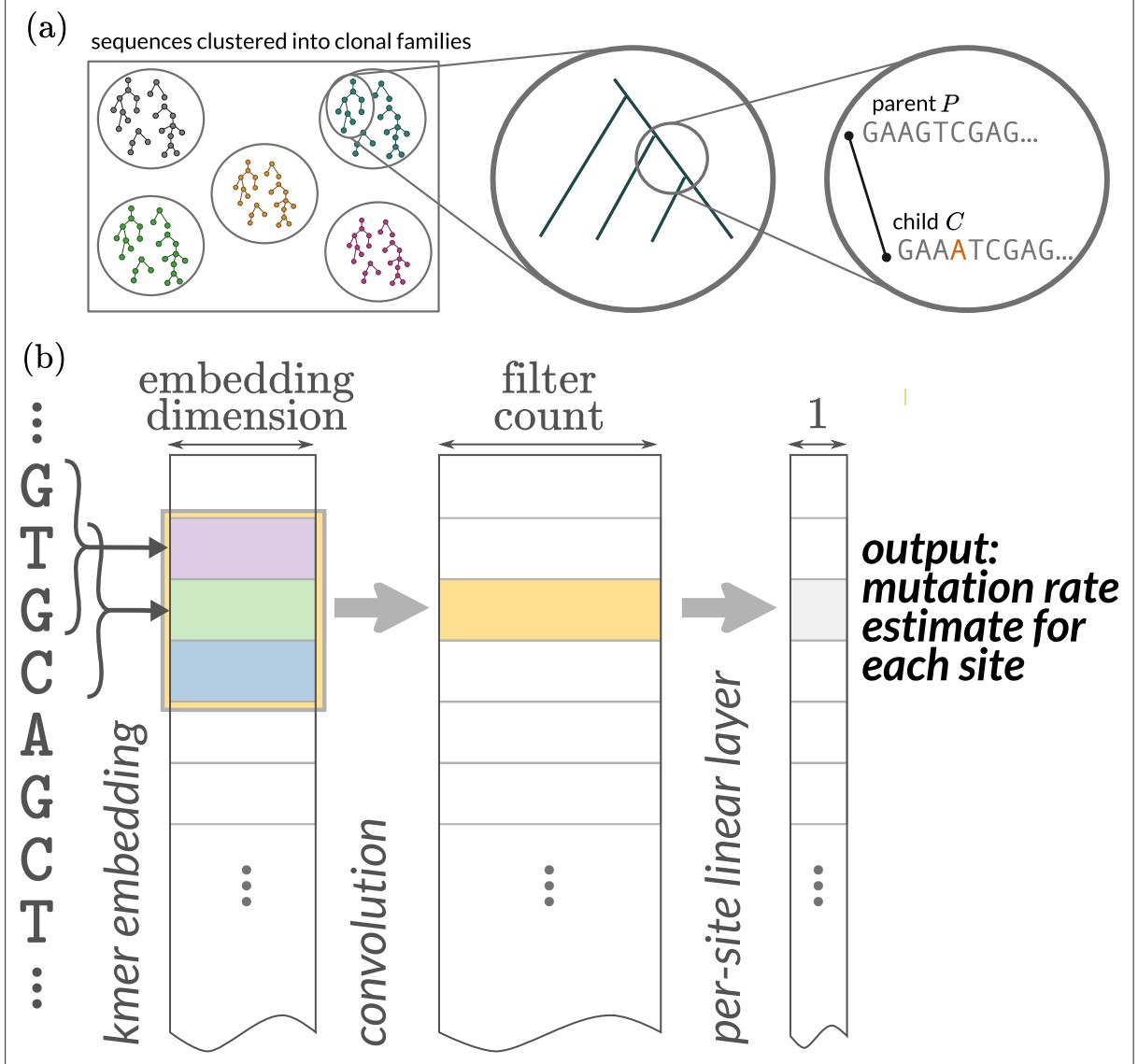

**Figure 1.** Overview of data processing and objective. (**a**) Out-of-frame sequences are clustered into clonal families. Trees are built on clonal families and then ancestral sequences are reconstructed using a simple sequence model. The prediction task is to predict the location and identity of mutations of child sequences given parent sequences. (**b**) Strategy for 'thrifty' convolutional neural networks with relatively few parameters. We use a trainable embedding of each 3-mer into a space; downstream convolutions happen on sequential collections of these embeddings. The 'width' of the k-mer model is determined by the size of the convolutional kernel, which in this cartoon is 3. This would give us effectively a 5-mer model because the 3-mer model adds one base on either side of a convolution of length 3. For the sake of simplicity, the probability distribution of the new base conditioned on there being a substitution (which we call the conditional substitution probability [CSP]) is not shown. The CSP output can emerge in several ways (*Figure 1—figure supplement 1*).

The online version of this article includes the following figure supplement(s) for figure 1:

**Figure supplement 1.** Strategies for estimating both per-site rate and conditional substitution probability (CSP).

sequence cannot code for a productive receptor. Because the data is out-of-frame, this means that the sequences under consideration are less likely to have undergone selective pressure in the germinal centers and instead provide more information about the SHM process. We also provide more relevant parent sequences and predict finer-scale events by using phylogenetic reconstruction and ancestral sequence inference on sequences clustered into clonal families (*Figure 1a*). We split the tree with ancestral sequences into pairs of parent and child sequences, which we call parent–child pairs. We also experiment with using synonymous mutation data by masking mutations from the loss function that are not synonymous (below; details in 'Materials and methods').

In all models, the distribution of mutations at a particular site is assumed to be independent of mutations at all other sites (but not independent of context). We follow many authors starting from *Yaari et al., 2013* in estimating a per-site rate, as well as a per-site probability distribution among the non-identical bases describing the base selected in the event of a mutation. We will call this the conditional substitution probability (CSP). For each site $i$, we assume that the mutation process is an exponential waiting time process with rate $\lambda_i$. Once the mutation occurs, we assume that the base is selected according to a categorical distribution with probabilities $\mathbf{p}_i$. Similar assumptions have been made previously (*Levinstein Hallak and Rosset, 2022*; *Levinstein Hallak et al., 2018*; *Rosset, 2007*; *Spisak et al., 2020*). To accommodate evolutionary time in our model, we include *offsets* in our exponential model—if $t$ is a branch length parameter for a sequence pair, we use parameter $\tilde{\lambda} = t\lambda$ for model inference so that the model is able to learn $\lambda$ irrespective of evolutionary time on a particular branch. This parameter $t$ is frequently the normalized mutation count (*Spisak et al., 2020*) but can be optimized as part of a joint optimization.

We used two data sets, which we will call the `briney` and `tang` data sets. The `briney` data (*Briney et al., 2019*) consists of samples from nine individuals, but two of these samples resulted in many more sequences than the rest. Thus, we will use a test–train split in which these two samples form the training data and the other seven samples form the testing data. We acknowledge the important work the *Spisak et al., 2020* team did in processing the `briney` data. The `tang` data (*Vergani et al., 2017*; *Tang et al., 2020*) will be a further test set. Details on data processing appear in 'Materials and methods'. In 'Materials and methods', we describe our attempts to find additional data sets.

## Models

We use the following strategy to combine the predictive power of local-context models without having the parameter penalty (*Figure 1b*). Each 3-mer is mapped into an embedding space of a fixed dimension, and these embedding locations are trainable parameters of the model. The idea is that the embedding abstracts some SHM-relevant characteristics of that 3-mer. Each sequence is then represented as a matrix with (sequence length) rows and (embedding dimension) columns. We then apply convolutional filters to these matrices, with taller convolutional filters effectively increasing the context of the model. For example, a kernel size of 11 gives effectively a 13-mer model (because of the additional base on either side of the 3-mer). We then apply a simple linear layer to the result of this step in order to get a mutation rate estimate for each site.

As described above, this class of models predicts both the per-site rate of SHM as well as the probability of alternate bases after mutation (called the CSP as above). We make these two model outputs in three ways (*Figure 1*, *Figure 1—figure supplement 1*): they can share everything except for the final layer ('joined' model), or they can share the embedding layer ('hybrid' model), or they can be estimated separately ('independent' model). A key difference with a full k-mer model is that when we increase the size of the kernel, the number of parameters increases linearly, not exponentially. In this way, the thriftiest well-performing model is effectively a 13-mer model with fewer parameters than a 5-mer model; however, one can scale these models among a variety of dimensions (*Table 1*).

**Table 1.** Selected model shapes and dropout probabilities.
The release name of the model is the name of the trained model released in the GitHub repository. The paper name is the name of the model used in this article, which describes more about its architecture. 'Kernel': the size of the convolutional kernel used in the model. 'Embed': the size of the embedding used for each 3-mer. Because there is one additional base on either side of a 3-mer, a model with kernel size 9 is effectively an 11-mer model, and a model with kernel size 11 is effectively a 13-mer model. The 'Medium' and 'Large' labels in the paper name designate the settings for Kernel, Embed, Filters, and Dropout.

| *Release name*: paper name | Kernel | Embed | Filters | Dropout | Params |
|---|---|---|---|---|---|
| *ThriftyHumV0.2-20*: CNN Joined Large | 11 | 7 | 19 | 0.3 | 2057 |
| 5mer | - | - | - | - | 3077 |
| Spisak | - | - | - | - | 3576 |
| *ThriftyHumV0.2-45*: CNN Indep Medium | 9 | 7 | 16 | 0.2 | 4539 |
| *ThriftyHumV0.2-59*: CNN Indep Large | 11 | 7 | 19 | 0.3 | 5931 |

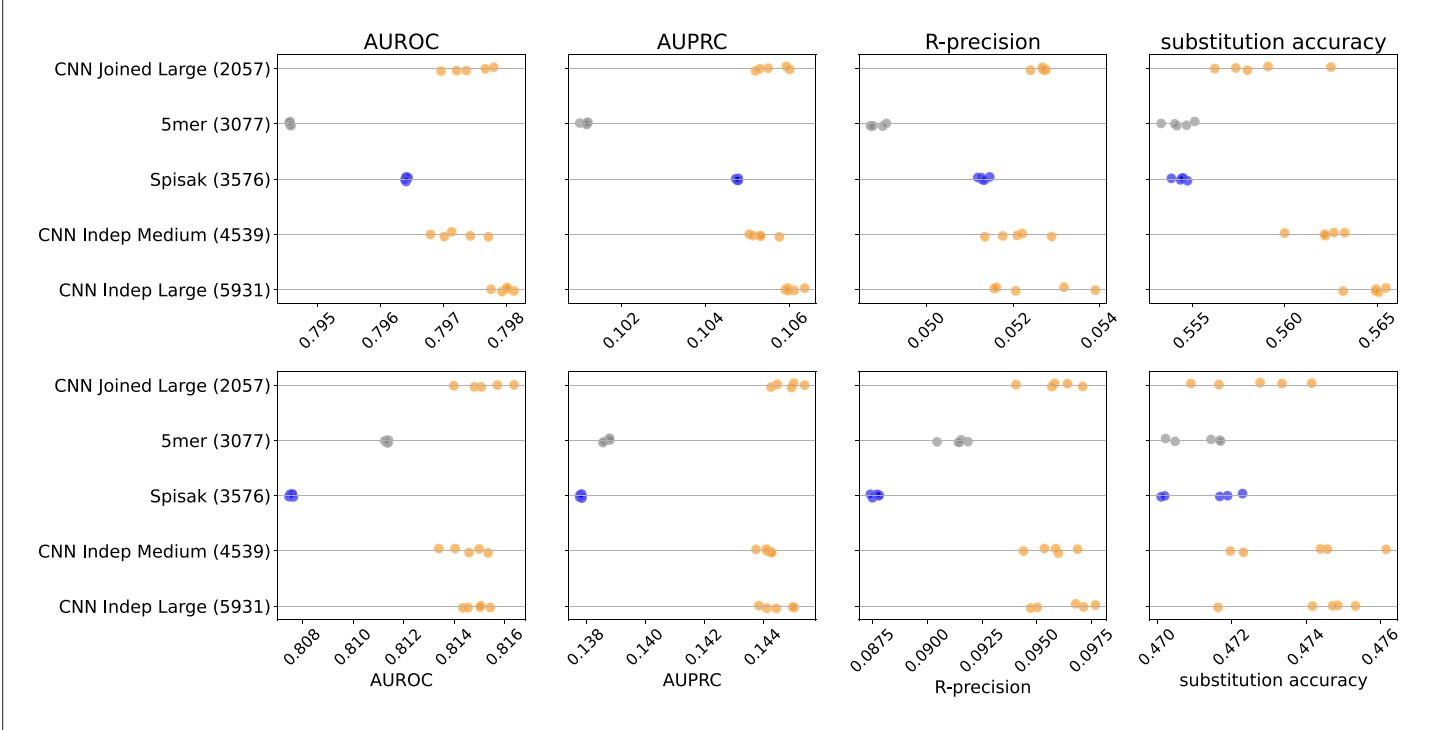

**Figure 2.** Model predictive performance on held-out samples: held-out individuals from the `briney` data (upper row) and on a separate sequencing experiment (`tang` data, lower row). Note that the two rows use different x-axis scales. The integer in parentheses indicates the number of parameters of the model. Each model has multiple points, each corresponding to an independent model training. This is a subset of the models for clarity; see *Figure 2—figure supplement 1* for all models.

The online version of this article includes the following figure supplement(s) for figure 2:

**Figure supplement 1.** Performance results for all the models.

**Figure supplement 2.** Agreement between the original 'shmoof' model of *Spisak et al., 2020* and our 'reshmoof' reimplementation.

**Figure supplement 3.** Performance comparison including S5F model with original coefficients.

We implemented our models in PyTorch (*Paszke, 2019*). Because of the small size of these models, they are fast to train and use. The hyperparameters for the models (*Table 1*) were selected with a run of Optuna (*Akiba et al., 2019*) early in the project and then fixed. Further optimization was not pursued because of the limited performance differences between the existing models.

## Thrifty CNNs give a modest performance improvement

In order to evaluate our proposed methods and compare to previous work, we first characterized the models in terms of predictive performance using area under the ROC curve (AUROC), area under the precision-recall curve (AUPRC), R-precision, and substitution accuracy. AUROC can be interpreted as the probability that the model correctly identifies sites that mutate as having higher mutability than those that do not. In fact, if one randomly selects a positive–negative pair, the AUROC is the probability that the positive example is assigned a higher probability than the negative example. However, this measure is sensitive to class imbalance, and we are in the imbalanced setting here because mutations are relatively rare. AUPRC provides an alternative that is less sensitive to class imbalance effects (*Ozenne et al., 2015*; *Saito and Rehmsmeier, 2015*) because the precision is the fraction of positive predictions that are true positives. R-precision gives a sense of how accurate the model is among sites that are most mutable. Specifically, if a given pair of parent and child sequences had $R$ mutations, R-precision is the precision of the predictions of mutability at the $R$ sites that are ranked as being most mutable. To evaluate performance at predicting per-base substitution probabilities (given a mutation occurred), we report substitution accuracy: how frequently is the predicted-most-likely base the one to which a site mutates?

We found that the thrifty convolutional neural network (CNN) models gave a modest performance improvement for these predictive metrics compared to existing models (*Figure 2, Figure 2—figure supplement 1*). Specifically, we compared to a 5-mer model trained in exactly the same way, as well as a reimplementation of the model of *Spisak et al., 2020*. We confirmed that our reimplementation infers very similar parameters to the previous implementation, although we add a slight regularization to avoid some aspects of the original model fit that appear to be artifacts (*Figure 2—figure supplement 2*). Because the *Spisak et al., 2020* model fits a per-site rate, and the `briney` data does not have full sequence coverage, we limited all evaluation to a region well covered by the `briney` data: positions 80 to 319, inclusive.

We were surprised to find that all metrics except substitution accuracy were better on data from a distinct sequencing experiment (`tang` data) than on held-out samples from the same sequencing experiment (`briney` data). We attribute this to there being a difference in sequencing error between the two experiments. The `briney` data allowed sequences with only a single UMI representative (*Spisak et al., 2020*) while the `tang` data required at least two sequences per UMI (*Tang et al., 2020*). If our model is successfully learning substitution probabilities due to SHM rather than the sequencing error, we would expect our model to underestimate the number of substitutions at positions with a low probability of substitution in the `briney` data (which we expect to have more sequencing error) but not in the `tang` data (which we expect to have less sequencing error). This is in fact what we see in our characterization of model fit below. We were also surprised to find that the per-site rate did not seem to help the 5-mer model on held-out data, despite the results of *Spisak et al., 2020*; see 'Discussion'.

We did not see a substantial performance improvement by increasing the number of parameters of the CNNs. Recall that our models differ in terms of how much they share between predicting the position of mutations and predicting their base identity (*Figure 1—figure supplement 1*). Although the 'Large Indep' model that has the most flexibility may do a slightly better job with held-out samples from the same experiment in terms of substitution accuracy, it does not appear any better when considering data from another experiment.

We also compared our work to a previous deep neural network model of SHM (*Tang et al., 2022*), which was trained on the `tang` data set. A DeepSHM model consists of a pair of CNN models, one for estimating mutation frequencies and another for CSPs; this is akin to our 'independent' model configuration. Each of these CNNs has over 250,000 parameters, so in total is about 100 times larger than the largest CNN model we trained. These CNNs are also slow to evaluate: because they make predictions one k-mer at a time, one must iterate over the sequence and obtain predictions for every site. Because the DeepSHM model cannot handle ambiguous nucleotides, we had to remove these from the evaluation. The authors found the best performance is achieved with 15-mers, which is comparable with the 11-mers and 13-mers in our thrifty models. We evaluated the DeepSHM 15-mer

**Table 2.** Performance evaluation on held-out `briney` data for S5F, DeepSHM, and thrifty models. The * on S5F indicates that this model was trained using synonymous mutations on a distinct data set to those considered here. The †on `tang` † is to signify that this is the tang data but with a different preprocessing scheme (*Tang et al., 2022*).

| Model | Training data | AUROC | AUPRC | R-prec | Sub. acc. |
|---|---|---|---|---|---|
| S5F | * | 0.775 | 0.0698 | 0.0290 | 0.514 |
| CNN Joined Large | `tang` | 0.787 | 0.0850 | 0.0406 | 0.510 |
| CNN Indep Medium | `tang` | 0.786 | 0.0848 | 0.0407 | 0.528 |
| CNN Indep Large | `tang` | 0.786 | 0.0852 | 0.0406 | 0.524 |
| DeepSHM | `tang`† | 0.786 | 0.0876 | 0.0421 | 0.537 |
| CNN Joined Large | `tang+briney` | 0.793 | 0.0923 | 0.0439 | 0.551 |
| CNN Indep Medium | `tang+briney` | 0.793 | 0.0919 | 0.0441 | 0.560 |
| CNN Indep Large | tang+briney | 0.794 | 0.0926 | 0.0450 | 0.562 |

AUPRC, area under the precision-recall curve; AUROC, area under the ROC curve; R-prec, R-precision; sub. acc., substitution accuracy.

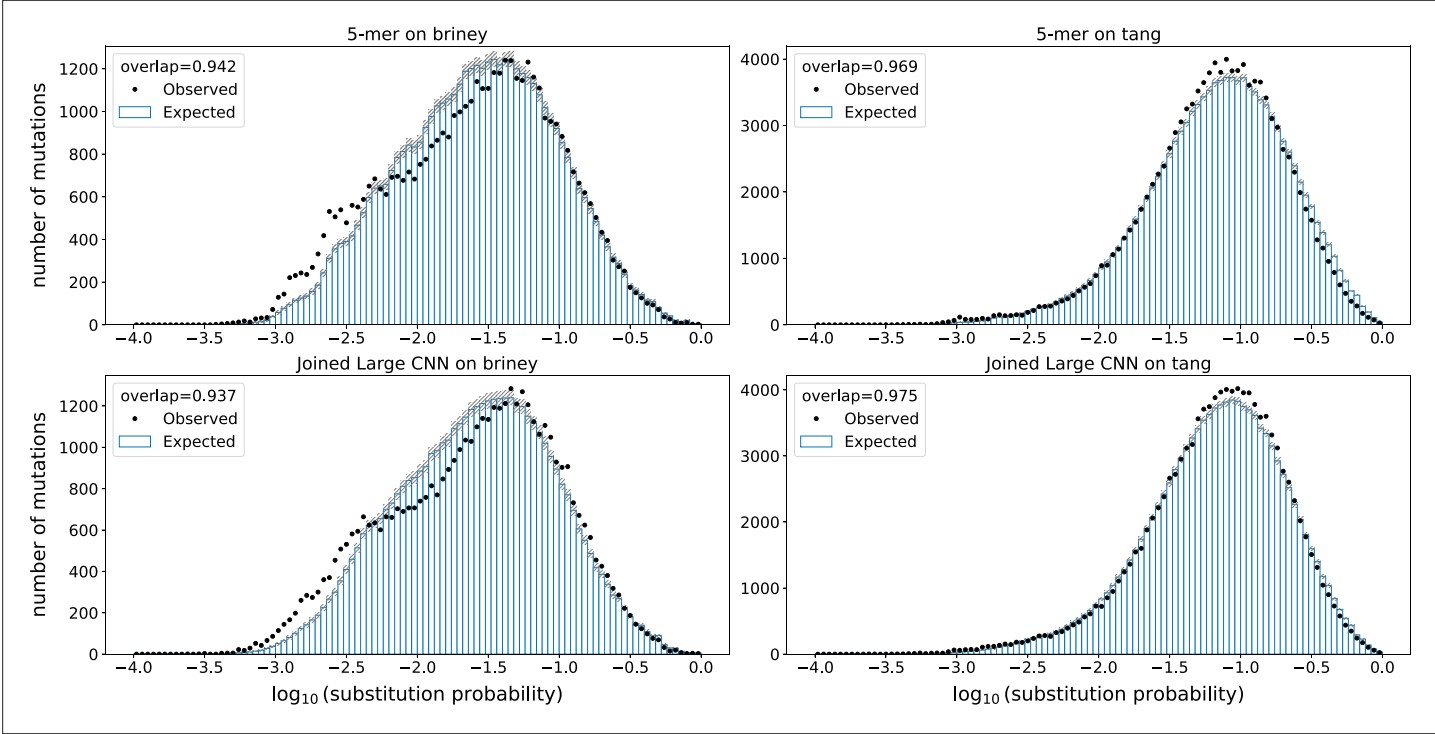

**Figure 3.** Model fit on held-out samples: held-out individuals from the `briney` data (upper row) and on a separate sequencing experiment (`tang` data, lower row). Observed mutations in held-out data are placed in bins according to their probability of mutation. For every bin, the points show the observed number of mutations in that bin, while the bars show the expected number of mutations in that bin. The overlap metric is the area of the intersection of the observed and expected divided by the average area between the two.

The online version of this article includes the following figure supplement(s) for figure 3:

**Figure supplement 1.** Comparison of held-out log likelihoods between the models.

model on the subset of the `briney` data and found it performs better than S5F but comparable to our models (*Table 2*). Specifically, our models performed comparably when trained on the `tang` data only, but when trained on the combined `tang` data and `briney` two largest repertoires, our models performed slightly better on the `briney` held-out repertoires.

We next characterized models in terms of out-of-sample model fit. For each site in the parent of each parent–child pair (PCP) of sequences, we computed the probability of a nucleotide substitution at that site in the corresponding child. We then compared the sum of those probabilities to the actual number of substitutions that were observed at each site in the PCPs (*Figure 3*). If the observed and expected counts match, then the model is doing a good job, on average, of predicting site-specific probabilities of substitution. We assessed matching using an 'overlap' metric, which quantifies the size of the intersection of the histograms divided by the average area of the histograms. We also assessed model log likelihood. These assessments were performed after branch length optimization to maximize likelihood.

We found that, as with the predictive metrics, all models gave similar performance, the differences of which were smaller than the differences between data sets. The overlap metric was better on the 5-mer model for the held-out `briney` data, but was better for the CNN models for the `tang` data. The log likelihood was slightly better for the CNN model for all held-out data sets (*Figure 3—figure supplement 1*).

## Further model elaborations did not improve out of sample performance

We tried adding a per-site rate to our CNN models, as well as other elaborations such as a transformer model directly on the amino acid embeddings, and a transformer combined with a CNN. We also tried adding a positional encoding to the input for the CNN model, which does not require additional

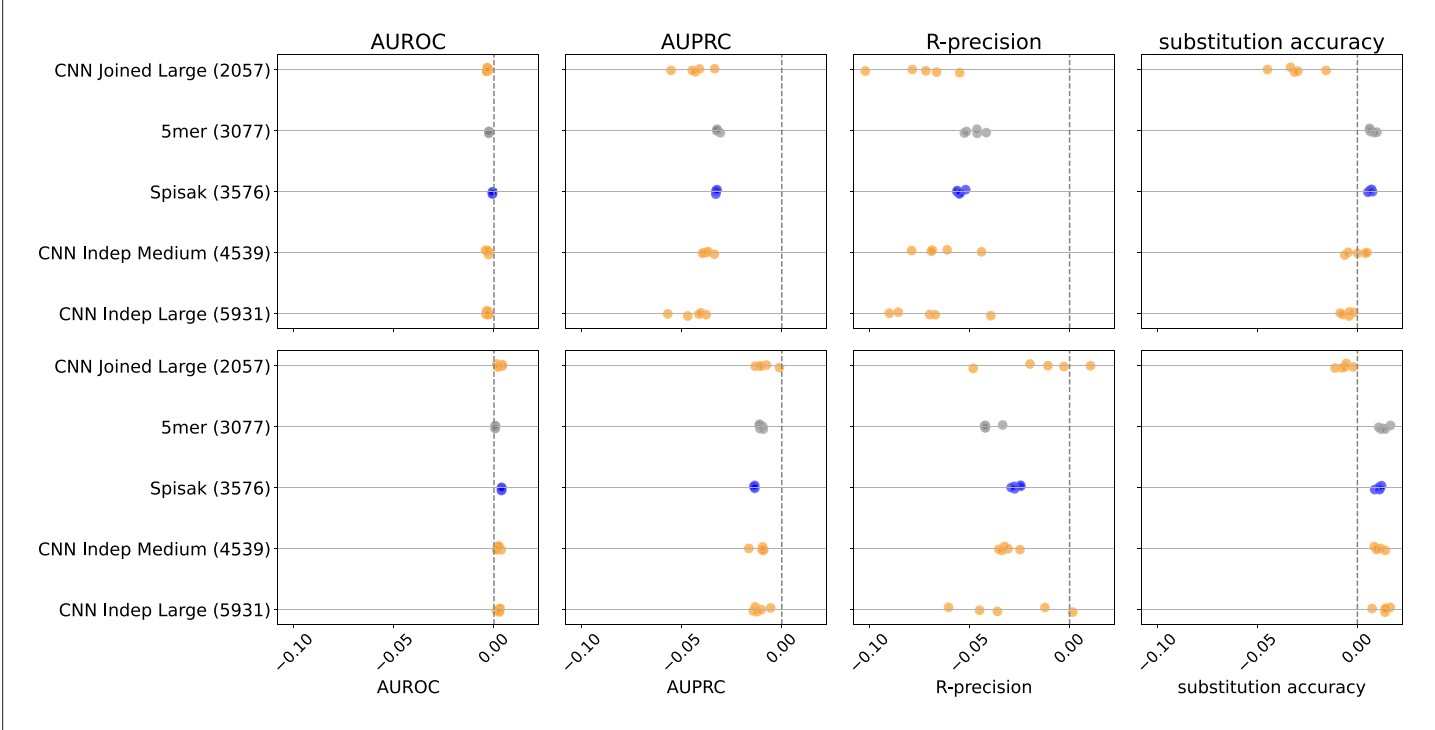

**Figure 4.** The relative change in performance for each statistic, namely the statistic for the model trained with out-of-frame (OOF) data and synonymous data, minus the statistic for the model trained with OOF data only, divided by the statistic trained with OOF data only. Thus, adding in synonymous mutations to the training set does not help predict on held-out out-of-frame data. Results shown for `briney` data (upper row) and `tang` data (lower row).

The online version of this article includes the following figure supplement(s) for figure 4:

**Figure supplement 1.** S5F performs better for mutation position prediction on synonymous mutations in the `jaffe` data than any model trained on out-of-frame data.

**Figure supplement 2.** S5F performs better for mutation position prediction on synonymous mutations in the `tang` data than any model trained on out-of-frame data.

parameters, but rather perturbs the input embeddings in a way that indicates their position in the sequence (***Vaswani et al., 2017***). None of these elaborations improved performance on held-out data. All of these experiments can be found in notebooks in the GitHub repository associated with this article.

We also found that jointly optimizing branch lengths along with model parameters did not improve out-of-sample performance.

## Out-of-frame evolution and synonymous mutations give different results

We conclude from the above that the richness of these models is limited by data volume, but unfortunately we were not able to find additional data sets with many out-of-frame sequences (see end of 'Materials and methods'). This raised the question of if we can use synonymous mutations of productive sequences to augment our data set. To do so, we trained models using a combination of the original training using the `briney` data but also with the `tang` data where the loss function was restricted to fourfold synonymous sites.

We found that adding these synonymous mutations to the training set reduced performance on the held-out out-of-frame data (***Figure 4***). Furthermore, when we train in the usual way using the `briney` data and evaluate on synonymous data, we do significantly worse than the S5F model, which was trained on synonymous mutations (***Figure 4***, ***Figure 4—figure supplements 1 and 2***). Our conclusion is that these two sources of data capture the effect of different processes.

## Discussion

Models of SHM are important to understand affinity maturation of B cells. They can also provide insight into the mechanism of SHM itself. Here we have developed a collection of models and worked to learn what they can teach us about SHM.

Specifically, by using an overlapping 3-mer embedding approach, we can parameterize wide-context models with relatively few parameters. For example, we can parameterize a 13-mer model with fewer parameters than a 5-mer model with better out-of-sample performance. A similar approach of embedded k-mers has been successful previously for genome regulatory element function prediction (*Ji et al., 2021*).

Our first main result is that these models are better than previous models using a 5-mer context, but only slightly so. This is interesting given that more distant effects are well motivated biologically, both in terms of the consensus model of SHM involving DNA damage, stripping, and repair (*Pilzecker and Jacobs, 2019*), as well as more recent results showing that the mesoscale environment of the BCR is important for SHM (*Wang et al., 2023*). Although another model formulation may be able to pick up these features, the present approach should be able to pick up features such as a nearby AID hotspot.

We found that adding a per-site rate to our models did not help predict on out-of-sample data. This is in contrast to recent work of *Spisak et al., 2020* suggesting that a per-site rate was helpful to predict SHM. We suspect that this contrast may be because of the means of evaluating this model. The *Spisak et al., 2020* paper quantifies an improved model fit over a 5-mer model by calculating a Pearson $r^2$ for each region comparing the model prediction to the aggregated mutation count per site for sites in that region. The *Spisak et al., 2020* model is itself parameterized in a per-site way, and so it is not surprising that the model has excellent fit according to this metric (Figure 4J of *Spisak et al., 2020*). Although they did a data-splitting exercise, this data split was on the level of parent–child pairs (not specified in the original paper; clarified via personal communication), which means that many of the parent sequences in the training set were very similar to parent sequences (*Spisak et al., 2020*) and actually not to a separately trained 5-mer model, but to the 5-mer component $\gamma_w$ of the joint model (typo in the original paper; clarified via personal communication).

We tried other means of adding per-site rates to the model, such as positional encoding and a transformer component, but these did not help. Although it is quite possible that sites evolve differently according to their absolute position in the sequence, our work shows that this is not necessary as part of a predictive model. Due to commonalities between neighborhoods of sites in sequences, it is difficult or perhaps impossible to disentangle the effect of a per-site rate from the effect of the motif model given a sufficiently rich motif model.

From a biological perspective, our findings indicate that while per-site rate (*Cohen et al., 2011*; *Spisak et al., 2020*; *Zhou and Kleinstein, 2020*) and mesoscale-level (*Wang et al., 2023*) effects may exist in SHM, their impact appears insufficient to significantly enhance statistical modeling performance. We refrain from making further biological conclusions and suggest that more definitive conclusions require an explicitly mechanistic model (*Fisher et al., 2025*).

Our work also contrasts the two main ways of training a neutral model: one is to use synonymous mutations (*Yaari et al., 2012*), and the other is to use out-of-frame sequences (*Spisak et al., 2020*) or some other sequences that one can assume are evolving neutrally (*Cui et al., 2016*). Here we find evidence that these two methods give different results and that models trained on one task do not do well on the other. This may be from synonymous mutations being under selection due to codon usage effects or could be from only a subset of motifs being possible to estimate from synonymous data (*Yaari et al., 2013*). Another possible explanation is that the spatial correlation of mutations (*Spisak et al., 2020*) leads to correlation in mutations at synonymous and non-synonymous sites: as an extreme example, a fourfold synonymous site could be next to a site under strong purifying selection, and if these two sites only mutate together this would effectively lead to purifying selection on the synonymous site. One could also imagine that the out-of-frame sequences are under selection such as having an insertion–deletion mutation that throws a sequence under selection out of frame, but this is mitigated by the ancestral sequence reconstruction and the removal of the naive sequence from the tree. From a model-fitting perspective, the contrast between these two objectives is disappointing because productive sequences are much more available than out-of-frame sequences.

Overall, we have presented and tested a variety of new models with test–train splits and found a slight improvement using parameter sparse or 'thrifty' CNNs. One interesting aspect of these models is that they allow for a wider k-mer context without having a parameter explosion. It is possible that the conclusions would differ if we had considerably more data, though despite our best efforts (see 'Materials and methods') we were unable to find a large additional volume of out-of-frame data. If and when additional data becomes available, our reproducible analysis can be used to evaluate these models on that data.

## Materials and methods
### Data sets and data processing

Our primary data set is the one introduced by *Spisak et al., 2020* in their recent work on modeling SHM. The data consists of human, out-of-frame IgH sequences sampled from several individuals (*Briney et al., 2019*), aligned using pRESTO (*Vander Heiden et al., 2014*). The resulting data comes to us as a collection of trees, one for each clonal family with at least six observed sequences in an individual, complete with (ancestral and observed) sequences and branch lengths, as well as a collection of metadata annotating, among other things, the identity and position of the V gene in the sequence. Starting from the clonal family clustering and multiple sequence alignments from *Spisak et al., 2020*, we remove sequences containing gaps to avoid ambiguity of site positions due to insertions or deletions. Resulting clonal families with less than six observed sequences are discarded. Phylogenetic inference in each clonal family is then redone with IQ-TREE (*Minh et al., 2020*). We apply the K80 (*Kimura, 1980*) substitution model and use the germline sequence as an outgroup, following the method of *Spisak et al., 2020*, and additionally allow for mutation rate heterogeneity among sites with a four-category FreeRate model *Soubrier et al., 2012*; *Yang, 1995*. The edges of the inferred phylogenetic trees are taken as parent–child pairs, except for the edge containing the naive sequence outgroup.

Additionally, we use the `tang` data set of human IgH sequences previously used for modeling SHM by *Tang et al., 2022*. This data set, originally generated by *Tang et al., 2020*; *Vergani et al., 2017*, includes full-length BCR repertoires from 21 individuals. We obtained preprocessed samples directly from the authors (*Tang et al., 2020*). We extracted the IgH sequences from marginal zone (MZ), memory (M), and plasma (PC) B cells and completed germline and clonal family inferences with `partis` (*Ralph and Matsen, 2016a*; *Ralph and Matsen, 2016b*; *Ralph and Matsen, 2019*). We keep clonal families of two or more sequences that consist of only out-of-frame sequences. In `partis`, a sequence is considered out-of-frame if either of the conserved codons for cysteine or tryptophan bounding the CDR3 is out-of-frame with the germline V gene. We then perform phylogenetic inference and ancestral sequence reconstruction for each clonal family using IQ-TREE as described above. The first site of the sequence is set to align with the start of the germline V gene; if necessary, nucleotides before the start of the V gene are truncated or sites at the 5′ end with missing reads are padded with 'N'.

**Table 3.** Data used in this article.

`briney` data is from *Briney et al., 2019* after processing done by *Spisak et al., 2020*. `tang` data is from *Vergani et al., 2017*; *Tang et al., 2020* and was sequenced using the methods of *Vergani et al., 2017*. Out-of-frame sequences from `briney` and `tang` are used. For productive sequences from `tang`, only fourfold synonymous sites are used. `jaffe` data is from *Jaffe et al., 2022* sequenced using 10X, where only fourfold synonymous sites of productive sequences are used. The 'read/cell' column is the sequencing depth listed as average number of reads per cell. 'Samples' is the number of individual samples in the data set; in these data sets, each sample is from a distinct individual. 'CFs' is the number of clonal families in the data set. 'PCPs' is the number of parent–child pairs in the data set. 'Median mutations' is the median number of mutations per PCP in the data set.

| Name | Type | Sequencing methods | Read/cell | Samples | CFs | PCPs | Median mutations |
|------|------|-------------------|-----------|---------|-----|------|------------------|
| briney | Out-of-frame | Bulk | ~1 | 9 | 3903 | 52,915 | 2 |
| tang | Out-of-frame | Single cell | ~40 | 21 | 3209 | 9984 | 7 |
| tang | Productive | Single cell | ~40 | 21 | 157,863 | 820,837 | 1 |
| jaffe | Productive | Single cell | ~5000 | 4 | 67,304 | 282,732 | 1 |

A small number of the edges have very long branch lengths, some of which seem to correspond to improper alignments. Following the lead of the authors of *Spisak et al., 2020*, we filter our data to only include edges with fewer than 10 mutations, which results in a modest ~6.4% reduction in the available data.

The `jaffe` data set (*Jaffe et al., 2022*) consists of paired heavy chain and light chain, full-length sequences of productive antibodies from four donors. We process the data with `partis` and IQ-TREE, similar to the `tang` data set. The `jaffe` synonymous data set are parent–child pairs from `jaffe` where we mask sites in the child sequence that are not fourfold degenerate in the parent context. For each parent–child pair, we check each codon context in the parent sequence for fourfold degenerate sites. For these sites, the corresponding nucleotide in the child sequence is preserved while all other sites are masked.

The data sets used in this article are summarized in *Table 3*.

## Models

In all the models we propose, we model SHM using a two-part model. First, we assume that the occurrence of point mutations follows a per-site exponential waiting time that is dependent entirely on the parent sequence. Furthermore, we assume that all mutations occur simultaneously, so that we can ignore how the order of point mutations along a branch affects likelihood computations. The other output of the model is the prediction of base identity. We interpret the normalized version of these rates as conditional probabilities given that the mutation has occurred.

For the 'thrifty' models, we have an embedding of each 3-mer along the sequence, which is then the input for a convolutional layer (*Figure 1b*). The output of the convolutional layer is then the input for the mutation rate estimate, which is shown in *Figure 1b* as well as the CSP (*Figure 1—figure supplement 1*).

All models are implemented in PyTorch (*Paszke, 2019*) and can be found in the GitHub repository associated with this article.

## Model training

Our loss functions are the likelihood of child mutation locations using offset, as well as a categorical cross-entropy loss for the base identity. We sum these log losses together using a weight of 0.01 on the cross-entropy loss to approximately even out the contributions of these two sources of loss.

Models were trained for 100 epochs, with the Poisson offset simply being the normalized count of mutations in the child sequence. We also tried a more sophisticated approach of joint optimization of branch length and model parameters.

## Model evaluation

### AUROC

The receiver operating characteristic (ROC) curve is a method of visualizing the tradeoff between true-positive rate (TPR = $\frac{TP}{TP+FN}$) and false-positive rate (FPR = $\frac{FP}{TN+FP}$) as we vary the cutoff value we use for separating positive and negative predictions. Notice that the denominators in each statistic depend only on the true labels, so the tradeoff parallels the one between true positives and false positives. Given a random classifier, the TPR is expected to be equal to FPR, meaning that the ROC would be a diagonal line from (0, 0) to (1, 1).

In order to reduce the ROC curve to a single quantity for performance assessment, one can compute the AUROC, which, when compared to 0.5, gives a sense of how well the classifier is doing when compared with a random classifier. *Hanley and McNeil, 1982* show that AUROC is equivalent to the probability that the classifier correctly ranks a randomly chosen pair of points, one from the negative and one from the positive class.

### AUPRC

Class imbalance, where we have many more negative examples than positive examples, can muddle performance evaluation of a classifier. Specifically, a large number of correctly identified negative examples can obscure the relatively poor performance of a classifier on a class that is under-represented in the data. Another approach is to instead consider precision and recall (Saito and Rehmsmeier, 2015). Analogous to the ROC curve, the principle here is to track the precision ($\frac{TP}{TP+FP}$) and recall ($\frac{TP}{TP+FN}$) as

we vary the cutoff parameter and plot the resulting points. Similar to before, we can distill the information of this curve down to a single number in the unit interval by computing the AUPRC. The precision of a classifier that uniformly assigns sites to the positive and negative classes will be

$$\rho = \frac{\text{\# mutated sites}}{\text{\# sites}}$$

in expectation. Thus, if we exclude pathologically bad classifiers, $\rho$ forms a baseline minimum value of the AUPRC. In contrast with the AUROC, neither precision nor recall is affected by the addition or removal of true negative (i.e., non-mutated sites that were predicted not to mutate) examples, and in fact the relationship is driven by the tradeoff between false positives and false negatives.

### R-precision

In order to characterize the ability of our classifiers to correctly identify sites that will mutate, we consider another metric: R-precision. This is the easiest to explain by first introducing top-$k$ precision, in which we take the $k$ 'hottest' (i.e., predicted to mutate with the highest probability) sites according to our classifier and compute the precision on those sites. We can refine top-$k$ precision above to $R$-precision, which is the top-$k$ precision when $k$ is set equal to the expected or observed number of mutations. This value can be interpreted in the following way: an $R$-precision of 0.1 means that if the classifier is told to predict the correct sites to mutate given their count, 10% of them will have actually mutated. As with AUPRC, a random classifier will have an $R$-precision of $\rho$.

### Substitution accuracy

As described above, when evaluating performance at predicting per-base substitution probabilities (given a mutation occurred), we report accuracy: how frequently is the predicted-most-likely base the one to which a site mutates?

## Software

Our work is released in an open-source Python package https://github.com/matsengrp/netam (*Matsen and Dumm, 2025*), with a simple API that makes it easy to train and evaluate models. We release trained models and their weights. Our analysis for this article is reproducible via https://github.com/matsengrp/thrifty-experiments-1(copy archived at *Sung, 2025*), which includes notebooks that reproduce the figures and tables in this article. We used the following software: PyTorch (*Paszke, 2019*), pandas (*McKinney, 2010*), matplotlib (*Hunter, 2007*), seaborn (*Waskom, 2021*), snakemake (*Mölder, 2021*), pytest (*Krekel et al., 2004*), and biopython (*Cock et al., 2009*).

## Attempts to find additional full-length, out-of-frame sequences

While the primary data set used here and originally by *Spisak et al., 2020* provides a large number of out-of-frame human IgH sequences, the sequencing methods used resulted in minimal coverage at the start of the V gene and thus limited information for that region (*Briney et al., 2019*). Alternatively, the Tang data set provides relatively high coverage along the full IgH sequence, but is limited in the amount of unique sequences sampled. We sought to supplement this data with additional full-length, out-of-frame human IgH sequences. Full-length IgH data is generally limited by design in common sequencing approaches. Even productive IgH data show low coverage at the start of the V gene, as evidenced by over 40% of the sequences in the Observed Antibody Space (OAS) database lacking sequence data for the first 15 amino acids (*Olsen et al., 2022b*; *Kovaltsuk et al., 2018*; *Olsen et al., 2022a*). We focus our efforts on data sets from recent studies focused on full-length antibody sequencing (*Ford et al., 2023*; *Soto et al., 2019*; *Rodriguez et al., 2023*). For all data sets, we implemented the `partis`-IQ-TREE pipeline as previously described on pre-processed data and extracted parent–child pairs for all clonal families of size 2+. Overall, out-of-frame sequences make up a relatively small proportion of sequence data (with this proportion varying by study/sequencing protocols), and none of the data sets considered had enough depth to extract a meaningful amount of out-of-frame sequence data for our purposes. The details of these efforts are described below.

From *Ford et al., 2023*, we obtained preprocessed data for 10 IgG FLAIRR-seq samples from the authors. Using consensus sequences from UMIs that were observed more than once (DUPCOUNT>1), we recovered 7938 out-of-frame sequences across all samples. These sequences belonged to 226

clonal families of size 2+ and 2633 singletons. This amounted to only 722 parent–child pairs, of which only 324 contained a mutation event. In attempts to extract more out-of-frame data, we additionally ran our pipeline on the preprocessed data including consensus sequences that were only observed once (and thus included no UMI error correction). From the 10 samples, we were able to recover 52,785 out-of-frame sequences, some of which were not UMI error corrected. This resulted in 2415 clonal families of size 2+ and 22,983 singletons, but still only gave us 6296 parent–child pairs with mutation events (9114 in total). We note that this sequencing method provided a higher proportion of out-of-frame sequences (9% for fully preprocessed data) than other methods we considered.

From *Rodriguez et al., 2023*, we obtained preprocessed 5' RACE AIRR-seq sequencing data for 51 IgG samples from the authors. We ran our pipeline on the preprocessed data and recovered only 4337 out-of-frame sequences in total. These sequences belonged to 188 clonal families of size 2+ and 1925 singletons. Of the 889 parent–child pairs from non-singletons, only 215 contained a mutation event. While the proportion of out-of-frame sequences was on par with the primary data set, sequence depth per sample was shallow and thus out-of-frame data was limited.

From *Soto et al., 2019*, we obtained preprocessed data for all three HIP donors from the authors. We ran our pipeline on a large subset of the data (sampling the first 1 million sequences for each donor IgH fasta file) to assess its potential for our purposes. From the 3 million sequences processed, we extracted 2686 out-of-frame sequences in total. These sequences corresponded to 11 clonal families of size 2+ and 2618 singletons. We obtained 102 parent–child pairs from non-singletons, of which only 57 contained a mutation event. The relatively low recovery of out-of-frame sequences in this subset of the data suggested that processing the full data set would not yield a meaningful amount of parent–child pairs for this study. We additionally observed that all of these sequences had no coverage at the start of the V gene, missing the first 12–60 bases.

## Acknowledgements

We thank the authors of *Spisak et al., 2020* for providing the data and answering questions about their work. We are also grateful to the authors of *Tang et al., 2022* for providing the data and answering questions about their work, as well as to the lab of Corey Watson for sharing data from *Ford et al., 2023* and *Rodriguez et al., 2023*. Additionally, we thank Luke Myers and the authors of *Soto et al., 2019* for sharing their preprocessed data. This work was supported by NIH grant R01-AI146028. Scientific Computing Infrastructure at Fred Hutch funded by ORIP grant S10OD028685. Frederick Matsen is an investigator of the Howard Hughes Medical Institute. This work was partially completed at the Kavli Institute for Theoretical Physics (KITP) at the University of California, Santa Barbara, and thus was supported by grant no. NSF PHY-2309135 to the Kavli Institute for Theoretical Physics (KITP) and the Gordon and Betty Moore Foundation Grant No. 2919.02.

## Additional information

### Funding

| Funder | Grant reference number | Author |
| --- | --- | --- |
| National Institutes of Health | R01-AI146028 | Mackenzie M Johnson<br>Julia Fukuyama<br>Frederick A Matsen |
| National Institutes of Health | S10OD028685 | Kevin Sung<br>Mackenzie M Johnson<br>Will Dumm<br>Noah Simon<br>Hugh Haddox<br>Julia Fukuyama<br>Frederick A Matsen |
| Howard Hughes Medical Institute | | Kevin Sung<br>Will Dumm<br>Hugh Haddox<br>Frederick A Matsen |

| Funder | Grant reference number | Author |
|--------|------------------------|--------|
| National Science Foundation | PHY-2309135 | Kevin Sung<br>Mackenzie M Johnson<br>Will Dumm<br>Noah Simon<br>Hugh Haddox<br>Julia Fukuyama<br>Frederick A Matsen |
| Gordon and Betty Moore Foundation | 2919.02 | Hugh Haddox<br>Frederick A Matsen |

The funders had no role in study design, data collection and interpretation, or the decision to submit the work for publication.

## Author contributions

Kevin Sung, Mackenzie M Johnson, Data curation, Software, Validation, Methodology, Writing – review and editing; Will Dumm, Software, Validation, Methodology, Writing – review and editing; Noah Simon, Hugh Haddox, Julia Fukuyama, Methodology, Writing – review and editing; Frederick A Matsen, Conceptualization, Software, Formal analysis, Funding acquisition, Validation, Visualization, Methodology, Writing – original draft, Project administration, Writing – review and editing

## Author ORCIDs

Kevin Sung https://orcid.org/0000-0002-7289-845X
Mackenzie M Johnson https://orcid.org/0000-0002-3915-2023
Julia Fukuyama https://orcid.org/0000-0002-7590-5563
Frederick A Matsen https://orcid.org/0000-0003-0607-6025

Reviewer #1 (Public review): https://doi.org/10.7554/eLife.105471.3.sa1
Reviewer #2 (Public review): https://doi.org/10.7554/eLife.105471.3.sa2
Reviewer #3 (Public review): https://doi.org/10.7554/eLife.105471.3.sa3
Author response https://doi.org/10.7554/eLife.105471.3.sa4

# Additional files

## Supplementary files
MDAR checklist

## Data availability

Processed data for this study is available on Dryad, https://doi.org/10.5061/dryad.np5hqc044. Software to produce the figures in the manuscript is available on GitHub (https://github.com/matsengrp/thrifty-experiments-1; copy archived at *Sung, 2025*).

The following previously published dataset was used:

| Author(s) | Year | Dataset title | Dataset URL | Database and Identifier |
|-----------|------|---------------|-------------|-------------------------|
| Matsen IV FA, Sung K, Johnson MM | 2025 | B cell receptor parent-child pairs for studying somatic hypermutation | https://doi.org/10.5061/dryad.np5hqc044 | Dryad Digital Repository, 10.5061/dryad.np5hqc044 |

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
