## [Editor Report · eLife Assessment]

This study provides an **important** method to model the statistical biases of hypermutations during the affinity maturation of antibodies. The authors show **convincingly** that their model outperforms previous methods with fewer parameters; this is made possible by the use of machine learning to expand the context dependence of the mutation bias. They also show that models learned from nonsynonymous mutations and from out-of-frame sequences are different, prompting new questions about germinal center function. Strengths of the study include an open-access tool for using the model, a careful curation of existing data sets, and a rigorous benchmark; it is also shown that current machine-learning methods are currently limited by the availability of data, which explains the only modest gain in model performance afforded by modern machine learning.

---

## [Referee Report · Reviewer #1 (Public review)]

Summary:

This paper introduces a new class of machine learning models for capturing how likely a specific nucleotide in a rearranged IG gene is to undergo somatic hypermutation. These models modestly outperform existing state-of-the-art efforts, despite having fewer free parameters. A surprising finding is that models trained on all mutations from non-functional rearrangements give divergent results from those trained on only silent mutations from functional rearrangements.

Strengths:

* The new model structure is quite clever and will provide a powerful way to explore larger models.

* Careful attention is paid to curating and processing large existing data sets.

* The authors are to be commended for their efforts to communicate with the developers of previous models and use the strongest possible versions of those in their current evaluation.

Weaknesses:

* No significant weaknesses noted

---

## [Referee Report · Reviewer #2 (Public review)]

This work offers an insightful contribution for researchers in computational biology, immunology, and machine learning. By employing a 3-mer embedding and CNN architecture, the authors demonstrate that it is possible to extend sequence context without exponentially increasing the model's complexity. Key findings include:

• Efficiency and Performance: Thrifty CNNs outperform traditional 5-mer models and match the performance of significantly larger models like DeepSHM.

• Neutral Mutation Data: A distinction is made between using synonymous mutations and out-of-frame sequences for model training, with evidence suggesting these methods capture different aspects of SHM, or different biases in the type of data.

• Open Source Contributions: The release of a Python package and pretrained models adds practical value for the community.

However, readers should be aware of the limitations. The improvements over existing models are modest, and the work is constrained by the availability of high-quality out-of-frame sequence data. The study also highlights that more complex modeling techniques, like transformers, did not enhance predictive performance, which underscores the role of data availability in such studies.

---

## [Referee Report · Reviewer #3 (Public review)]

Summary:

Modeling and estimating sequence context biases during B cell somatic hypermutation is important for accurately modeling B cell evolution to better understand responses to infection and vaccination. Sung et al. introduce new statistical models that capture a wider sequence context of somatic hypermutation with a comparatively small number of additional parameters. They demonstrate their model's performance with rigorous testing across multiple subjects and datasets. Prior work has captured the mutation biases of fixed 3-, 5-, and 7-mers, but each of these expansions has significantly more parameters. The authors developed a machine-learning-based approach to learn these biases using wider contexts with comparatively few parameters.

Strengths:

Well motivated and defined problem. Clever solution to expand nucleotide context. Complete separation of training and test data by using different subjects for training vs testing. Release of open-source tools and scripts for reproducibility.

The authors have addressed my prior comments.

---

## [Author Response]

The following is the authors’ response to the previous reviews

**Reviewer #1 (Public Review):**
Summary:This paper introduces a new class of machine learning models for capturing how likely a specific nucleotide in a rearranged IG gene is to undergo somatic hypermutation. These models modestly outperform existing state-of-the-art efforts, despite having fewer free parameters. A surprising finding is that models trained on all mutations from non-functional rearrangements give divergent results from those trained on only silent mutations from functional rearrangements.Strengths:(1) The new model structure is quite clever and will provide a powerful way to explore larger models.(2) Careful attention is paid to curating and processing large existing data sets.(3) The authors are to be commended for their efforts to communicate with the developers of previous models and use the strongest possible versions of those in their current evaluation.

Thank you very much for your comments. We especially appreciate the last comment, as we have indeed tried hard to do so.

Weaknesses:(1) 10x/single cell data has a fairly different error profile compared to bulk data. A synonymous model should be built from the same
briney
dataset as the base model to validate the difference between the two types of training data.

Thank you for pointing this out.

We have repeated the same analysis with synonymous mutations derived from the bulk-sequenced

The fact that both the 10x and the

(2) The decision to test only kernels of 7, 9, and 11 is not described. The selection/optimization of embedding size is not explained. The filters listed in Table 1 are not defined.

We have added the following to the Models subsection to further explain these decisions:

“The hyperparameters for the models (Table 1) were selected with a run of Optuna (Akiba et al., 2019) early in the project and then fixed. Further optimization was not pursued because of the limited performance differences between the existing models.”

**Reviewer #2 (Public Review):**
Summary:This work offers an insightful contribution for researchers in computational biology, immunology, and machine learning. By employing a 3-mer embedding and CNN architecture, the authors demonstrate that it is possible to extend sequence context without exponentially increasing the model's complexity.Key findings:(1) Efficiency and Performance: Thrifty CNNs outperform traditional 5-mer models and match the performance of significantly larger models like DeepSHM.(2)Neutral Mutation Data: A distinction is made between using synonymous mutations and out-of-frame sequences for model training, with evidence suggesting these methods capture different aspects of SHM or different biases.(3) Open Source Contributions: The release of a Python package and pre-trained models adds practical value for the community.

Thank you for your positive comments. We believe that we have been clear about the modest improvements (e.g., the abstract says “slight improvement”), and we discuss the data limitations extensively. If there are ways we can do this more effectively, we are happy to hear them.

**Reviewer #3 (Public Review):**
Summary:Sung et al. introduce new statistical models that capture a wider sequence context of somatic hypermutation with a comparatively small number of additional parameters. They demonstrate their model’s performance with rigorous testing across multiple subjects and datasets.Strengths:Well-motivated and defined problem. Clever solution to expand nucleotide context. Complete separation of training and test data by using different subjects for training vs testing. Release of open-source tools and scripts for reproducibility.

Thank you for your positive comments.

Weaknesses:This study could be improved with better descriptions of dataset sequencing technology, sequencing depth, etc.

We have added columns to Table 3 that report sequencing technology and depth for each dataset.

**Reviewer #1 (Recommendations for the Authors):**
(1) There seems to be a contradiction between Tables 2 and 3 as to whether the Tang et al. dataset was used to train models or only to test them.

Thank you for catching this. The "purpose" column in Table 3 was for the main analysis, while Table 2 is describing only models trained to compare with DeepSHM. Explaining this seems more work than it's worth, so we simply removed that column from Table 2. The dataset purposes are clear from the text.

(2) In Figure 4, I assume the two rows correspond to the Briney and Tang datasets, as in Figure 2, but this is not explicitly described.

Yes, you are correct. We added an explanation in the caption of Figure 4.

(3) Figure 2, supplement 1 should include a table like Table 1 that describes these additional models.

We have added an explanation in the caption to Table 1 that "Medium" and "Large" refer to specific hyperparameter choices. The caption to Figure 2, supplement 1 now describes the corresponding hyperparameter choices for "Small" thrifty models.

(4) On line 378 "Therefore in either case" seems extraneous.

Indeed. We have dropped those words.

(5) In the last paragraph of the Discussion, only the attempt to curate the Ford dataset is described. I am not sure if you intended to discuss the Rodriguez dataset here or not.

Thank you for pointing this out. We have updated the Materials and Methods section to include our attempts to recover data from Rodriguez et al., 2023.

(6) Have you looked to see if Soto et al. (Nature 2019) provides usable data for your purposes?

Thank you for making us aware of this dataset!

We assessed it but found that the recovery of usable out-of-frame sequences was too low to be useful for our analysis. We now describe this evaluation in the paper.

(7) Cui et al. note a high similarity between S5F and S5NF (r=0.93). Does that constrain the possible explanations for the divergence you see?

This is an excellent point.

We don't believe the correlation observed in Cui and our results are incompatible. Our point is not that the two sources of neutral data are completely different but that they differ enough to limit generalization. Also, the Spearman correlation in Cui is 0.86, which aligns with our observed drop in R-precision.

(8) Are you able to test the effects of branch length or background SHM on the model?

We're unsure what is meant by “background SHM.”

We did try joint optimization of branch length and model parameters, but it did not improve performance. Differences in clone size thresholds do exist between datasets, but Figure 3 suggests that tang is better sequence data.

(9) Would the model be expected to scale up to a kernel of, say, 50? Would that help yield biological insight?

We did not test such large models because larger kernels did not improve performance.

While your suggestion is intriguing, distinguishing biological effects from overfitting would be difficult. We explore biological insights more directly in our recent mechanistic model paper (Fisher et al., 2025), which is now cited in a new paragraph on biological conclusions.

**Reviewer #2 (Recommendations for the Authors):**
(1) Consider applying a stricter filtration approach to the Briney dataset to make it more comparable to the Tang dataset.

Thank you. We agree that differences in datasets are interesting, though model rankings remain consistent. We now include supplementary figures comparing synonymous and out-of-frame models from the

(2) You omit mutations between the unmutated germline and the MRCA of each tree. Why?

The inferred germline may be incorrect due to germline variation or CDR3 indels, which could introduce spurious mutations. Following Spisak et al. (2020), we exclude this branch.

Yes, singletons are discarded: ~28k in tang and ~1.1M in jaffe.

(3) Could a unified model trained on both data types offer further insights?

We agree and present such an analysis in Figure 4.

(4) Tree inference biases from parent-child distances may impact the results.

While this is an important issue, all models are trained on the same trees, so we expect any noise or bias to be consistent. Different datasets help confirm the robustness of our findings.

(5) Simulations would strengthen validation.

We focused on real datasets, which we view as a strength. While simulations could help, designing a meaningful simulation model would be nontrivial. We have clarified this point in the manuscript.

**Reviewer #3 (Recommendations for the Authors):**
There are typos in lines 109, 110, 301, 307, and 418.

Thank you, we have corrected them.